# Adaptive bias correction for improved subseasonal forecasting

**Soukayna Mouatadid** [1] ✉, **Paulo Orenstein** [2], **Genevieve Flaspohler** [3,4,5], **Judah Cohen** [6,7], **Miruna Oprescu**[8], **Ernest Fraenkel** [9] & **Lester Mackey** [10] ✉

Subseasonal forecasting—predicting temperature and precipitation 2 to 6 weeks ahead—is critical for effective water allocation, wildfire management, and drought and flood mitigation. Recent international research efforts have advanced the subseasonal capabilities of operational dynamical models, yet temperature and precipitation prediction skills remain poor, partly due to stubborn errors in representing atmospheric dynamics and physics inside dynamical models. Here, to counter these errors, we introduce an *adaptive bias correction* (ABC) method that combines state-of-the-art dynamical forecasts with observations using machine learning. We show that, when applied to the leading subseasonal model from the European Centre for Medium-Range Weather Forecasts (ECMWF), ABC improves temperature forecasting skill by 60–90% (over baseline skills of 0.18–0.25) and precipitation forecasting skill by 40–69% (over baseline skills of 0.11–0.15) in the contiguous U.S. We couple these performance improvements with a practical workflow to explain ABC skill gains and identify higher-skill windows of opportunity based on specific climate conditions.

Improving our ability to forecast both weather and climate is of interest to many sectors of the economy and government agencies, from the local to the national level. Weather forecasts 0–10 days ahead and climate forecasts seasons to decades ahead are currently used operationally in decision making, and the accuracy and reliability of these forecasts has improved consistently in recent decades[1]. However, many critical applications—including water allocation, wildfire management, and drought and flood mitigation—require subseasonal forecasts, with lead times beyond 10 days and up to a season[2,3]. Given the changing nature of the climate and the increasing frequency of extreme weather events, there is a social and scientific consensus regarding the importance and urgency of providing reliable subseasonal forecasts[4,5].

Subseasonal forecasting lies in a challenging intermediate domain between shorter-term weather forecasting (an initial-value problem) and longer-term climate forecasting (a boundary value problem). Skillful subseasonal forecasting requires capturing the complex dependence between local weather conditions, typically described by numerical weather models, and global climate variables, usually part of long-range seasonal forecasts[2]. The intertwined dynamics of initial-value prediction problems and boundary forcing phenomena led subseasonal forecasting to long be considered a predictability desert[6], more difficult than either short-term weather forecasting or long-term climate prediction. Recent studies, however, have highlighted important sources of predictability on subseasonal timescales, including oscillatory modes such as El Niño–Southern Oscillation and the

[1]Department of Computer Science, University of Toronto, Toronto, ON, Canada. [2]Instituto de Matemática Pura e Aplicada, Rio de Janeiro, Brazil. [3]nLine Inc., Berkeley, CA, USA. [4]Department of Electrical Engineering and Computer Science, Massachusetts Institute of Technology, Cambridge, MA, USA. [5]Department of Applied Ocean Science and Engineering, Woods Hole Oceanographic Institution, Falmouth, MA, USA. [6]Atmospheric and Environmental Research, Lexington, MA, USA. [7]Department of Civil and Environmental Engineering, Massachusetts Institute of Technology, Cambridge, MA, USA. [8]Department of Computer Science, Cornell University, Ithaca, NY, USA. [9]Department of Biological Engineering, Massachusetts Institute of Technology, Cambridge, MA, USA. [10]Microsoft Research New England, Cambridge, MA, USA. ✉e-mail: soukayna@cs.toronto.edu; lmackey@microsoft.com

Madden–Julian oscillation (MJO), large-scale anomalies in, e.g., soil moisture or sea ice, and external forcing[4,7]. These predictability sources are imperfectly understood and imperfectly represented in weather and climate models[8] and hence represent an opportunity for more skillful subseasonal forecasting.

The challenges of subseasonal forecasting are particularly apparent for precipitation forecasts[9]. Precipitation is governed by both macro-scale dynamical processes of the atmosphere and complex microphysical processes, some of which are still not fully understood[8]. In addition, precipitation is oftentimes a very local phenomenon, working over a much finer scale than the resolution employed by subseasonal dynamical models. This scale incompatibility, in concert with suboptimal process representation and incomplete process understanding, results in dynamical models falling short of predictability limits for forecasting precipitation[4,8]. Generating precipitation forecasts for longer seasonal horizons is an even more daunting task. In this case, dynamical models show considerable biases in precipitation and wind fields. These biases arise from the parameterization of key physical processes associated with deep convective cloud systems[10] and, when combined with chaotic dynamics and imperfectly represented sources of predictability, translate into rapidly decreasing skill for precipitation forecasts.

Bridging the gap between short-term and seasonal forecasting has been the focus of several recent large-scale research efforts to advance the subseasonal capabilities of operational physics-based models[9,11,12]. However, despite these advances, dynamical models still suffer from persistent systematic errors, which limit the skill of temperature and precipitation forecasts for longer subseasonal lead times. Low skill at these time horizons has a palpable practical impact on the utility of subseasonal forecasts for policy planners and stakeholders.

To counter the observed systematic errors of physics-based models on the subseasonal timescale, there have been parallel efforts in recent years to demonstrate the value of machine learning and deep learning methods for improved subseasonal forecasting accuracy[13–25]. While these works demonstrate the promise of learned models for subseasonal forecasting, they also highlight the complementary strengths of physics- and learning-based approaches and the opportunity to combine those strengths to improve forecasting skill[15,20–22].

To harness the complementary strengths of physics- and learning-based models, we introduce a hybrid dynamical-learning framework for improved subseasonal forecasting. In particular, we learn to adaptively correct the biases of dynamical models and apply our adaptive bias correction (ABC) to improve the skill of subseasonal temperature and precipitation forecasts. ABC is an ensemble of three low-cost, high-accuracy machine learning models introduced in this work: Dynamical++, Climatology++, and Persistence++. Each model trains only on past temperature, precipitation, and forecast data and outputs corrections for future forecasts tailored to the site, target date, and dynamical model. Dynamical++ and Climatology++ learn site- and date-specific offsets for dynamical and climatological forecasts by minimizing forecasting error over adaptively selected training periods. Persistence++ additionally accounts for recent weather trends by combining lagged observations, dynamical forecasts, and climatology to minimize historical forecasting error for each site. More details on each component model can be found in "Methods" section. Correction alone is no substitute for improved understanding and representation of predictability sources, and we therefore view ABC as a complement for improved dynamical modeling. Fortunately, as an adaptive correction, ABC automatically benefits from scientific improvements to its dynamical model inputs while learning to compensate for their residual systematic errors.

ABC can be applied operationally as a computationally inexpensive enhancement to any dynamical model forecast, and we use this property to substantially reduce the forecasting errors of eight operational dynamical models, including the state-of-the-art ECMWF model. ABC also improves upon the skill of classical and recently developed bias corrections from the subseasonal forecasting literature including quantile mapping[26–29], locally estimated scatterplot smoothing (LOESS)[27,30], and neural network[22] approaches. We couple these performance improvements with a practical workflow for explaining ABC skill gains using Cohort Shapley[31] and identifying higher-skill windows of opportunity[5] based on relevant climate variables. To facilitate future deployment and development, we release our ABC model and workflow code through the `subseasonal_toolkit` Python package.

## Results

### Improved precipitation and temperature prediction with adaptive bias correction

Figure 1 highlights the advantage of ABC over raw dynamical models when forecasting accumulated precipitation and averaged temperature in the contiguous U.S. Here, ABC is applied to the leading subseasonal model, ECMWF, to each of seven operational models participating in the Subseasonal Experiment [SubX[11]], and to the mean of the SubX models. Subseasonal forecasting skill, measured by uncentered anomaly correlation, is evaluated at two forecast horizons, weeks 3–4 and weeks 5–6, and averaged over all available forecast dates in the 4-year span of 2018–2021. We find that, for each dynamical model input and forecasting task, ABC leads to a pronounced improvement in skill. For example, when applied to the U.S. operational Climate Forecast System Version 2 (CFSv2), ABC improves temperature forecasting skill by 109-289% (over baseline skills of 0.08-0.17) and precipitation skill by 165–253% (over baseline skills of 0.05–0.07). When applied to the leading ECMWF model, ABC improves temperature skill by 60–90% (over baseline skills of 0.18–0.25) and precipitation skill by 40–69% (over baseline skills of 0.11–0.15). Moreover, for precipitation, even lower-skill models like CCSM4 have improved skill that is comparable to the best dynamical model after the application of ABC. Overall—despite significant variability in dynamical model skill—ABC consistently reduces the systematic errors of its input model, bringing forecasts closer to observations for each target variable and time horizon. Similar forecast improvement is observed when stratifying skill by season (see Supplementary Fig. S1).

In Supplementary Fig. S2, we compare ABC with three additional subseasonal debiasing baselines (detailed in "Methods" section): quantile mapping[26–29], LOESS debiasing[27,30], and a recently proposed neural network debiasing scheme trained jointly on temperature and precipitation inputs [NN-A[22]]. Given the same dynamical model inputs, ABC improves upon the skill of each baseline for each target variable and forecast horizon.

We next examine the spatial distribution of skill for CFSv2, ECMWF, and their ABC-corrected counterparts at three forecast horizons in Fig. 2. At the shorter-term horizon of weeks 1–2, both CFSv2 and ECMWF have reasonably high skill throughout the contiguous U.S. However, skill drops precipitously for both models when moving to the subseasonal forecast horizons (weeks 3–4 and 5–6). This degradation is particularly striking for precipitation, where prediction skill drops to zero or to negative values in the central and northeastern parts of the U.S. For temperature prediction, CFSv2 has a skill of zero across a broad region of the East, while ECMWF produces isolated pockets of zero skill in the west. At these subseasonal timescales, ABC provides consistent improvements across the U.S. that either double or triple the mean skill of CFSv2 and increase the mean skill of ECMWF by 40–90% (over baseline skills of 0.11–0.25). Similar improvements are observed when ABC is applied to the SubX multimodel mean (see Supplementary Fig. S3). In addition, common skill patterns across models are apparent that are consistent with higher precipitation predictability in the Western U.S. than in the Eastern U.S. and higher

temperature predictability on the coasts than in the center of the country.

Notably, ABC also improves over standard operational debiasing protocols (labeled `debiased CFSv2` and `debiased ECMWF` in Fig. 2), tripling the average precipitation skill of debiased CFSv2 and increasing that of debiased ECMWF by 70% (over a baseline skill of 0.11). As seen in Supplementary Fig. S4, ABC additionally improves upon the quantile mapping, LOESS, and neural network debiasing baselines, doubling the ECMWF precipitation skill of the best-performing baseline and improving the ECMWF temperature skill by 37% (over a baseline skill of 0.25).

A practical implication of these improvements for downstream decision-makers is an expanded geographic range for actionable skill, defined here as spatial skill above a given sufficiency threshold. For example, in Fig. 3, we vary the weeks 5–6 sufficiency threshold from 0 to 0.6 and find that ABC consistently boosts the number of locales with actionable skill over both raw and operationally debiased CFSv2 and ECMWF. We observe similar gains for weeks 3–4 in Supplementary Fig. S5 and for ABC correction of the SubX multimodel mean in Supplementary Fig. S6.

We emphasize that our results, like those of refs. 21,22,27–29, focus on improved deterministic forecasting: outputting a more accurate point estimate of a future weather variable. The complementary paradigm of probabilistic forecasting instead predicts the distribution of a weather variable, i.e., the probability that a variable will fall above or below any given threshold. Ideally, one would employ a tailored approach to probabilistic debiasing that directly optimizes a probabilistic skill metric to output a corrected distribution. However, there is a simple, inexpensive way to convert the output of ABC into a probabilistic forecast. Given any ensemble of dynamical model forecasts (e.g., the control and perturbed forecasts routinely generated operationally), one can train ABC on the ensemble mean, apply the learned bias corrections to each ensemble member individually, and use the empirical distribution of those bias-corrected forecasts as the probabilistic forecasting estimate. In Supplementary Figs. S7–S10, we present two standard probabilistic skill metrics—the continuous ranked probability score (CRPS) and the Brier skill score (BSS) for above normal observations[32], defined in Supplementary Methods—and observe that, for each target variable, forecast horizon, and season, ABC improves upon the BSS and CRPS of ECMWF, LOESS debiasing, and quantile mapping debiasing.

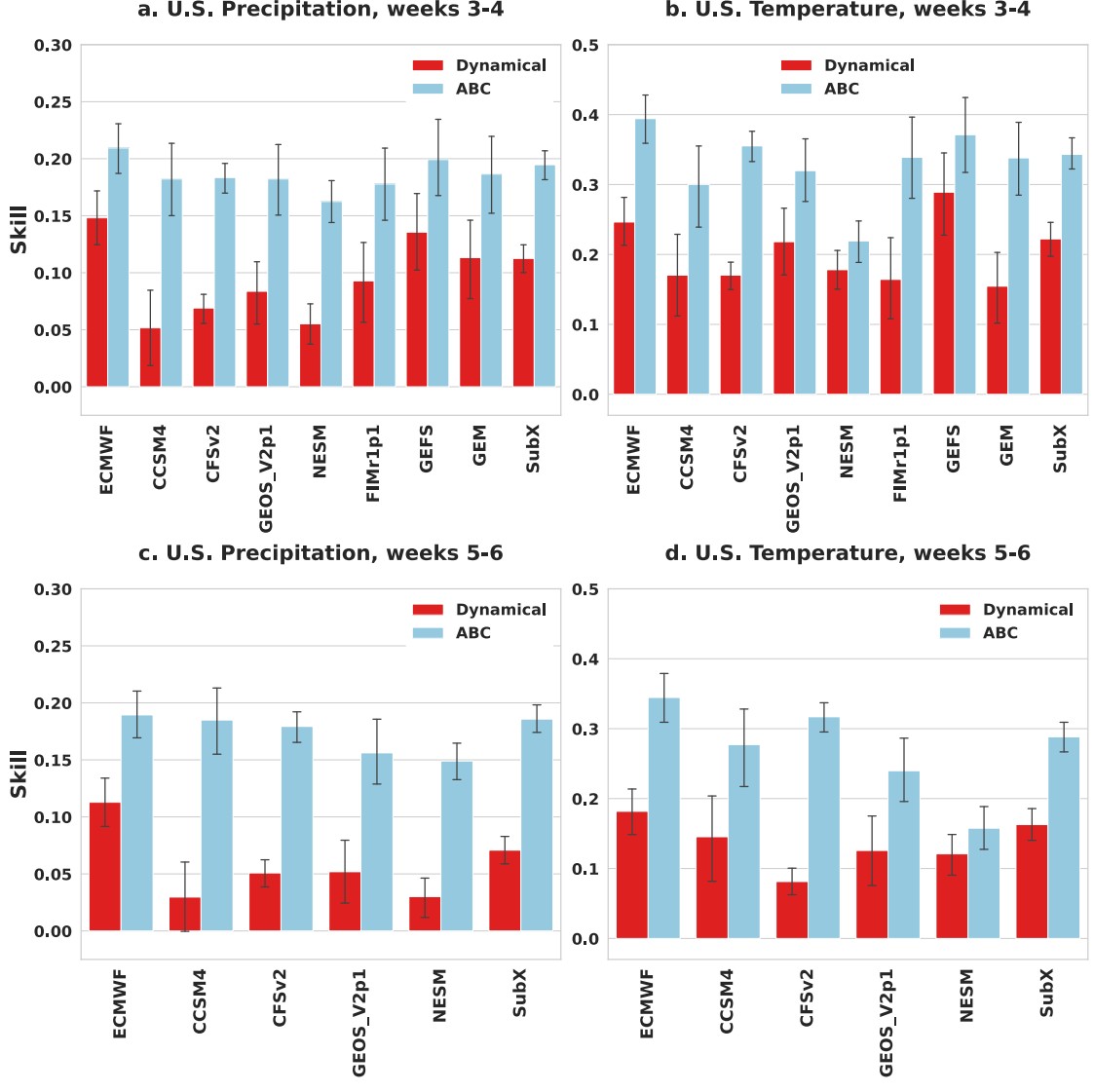

**Fig. 1 | Average forecast skill for dynamical models (red) and their adaptive bias correction (ABC) counterparts (blue).** Across the contiguous U.S. and the years 2018–2021, ABC provides a pronounced improvement in skill for each SubX or ECMWF dynamical model input and each forecasting task (**a–d**). The error bars display 95% bootstrap confidence intervals. Models without forecast data for weeks 5–6 are omitted from the bottom panels.

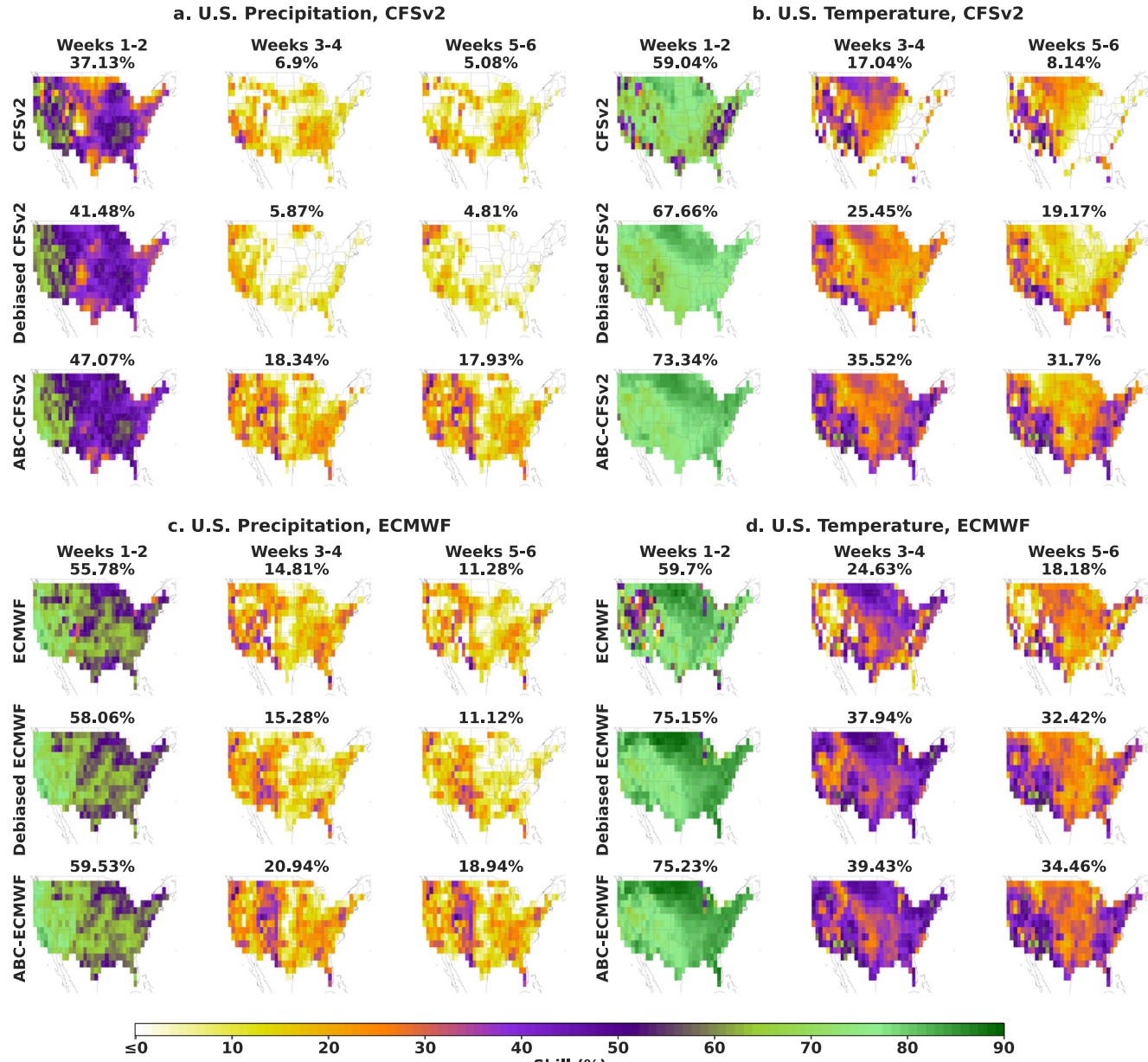

**Fig. 2 | Spatial skill distribution of dynamical models and their adaptive bias corrections.** Across the contiguous U.S. and the years 2018–2021, dynamical model skill drops precipitously at subseasonal timescales (weeks 3–4 and 5–6), but adaptive bias correction (ABC) attenuates the degradation, doubling or tripling the skill of CFSv2 (**a, b**) and boosting ECMWF skill 40–90% over baseline skills of 0.11–0.25 (**c, d**). Taking the same raw model forecasts as input, ABC also provides consistent improvements over operational debiasing protocols, tripling the precipitation skill of debiased CFSv2 and improving that of debiased ECMWF by 70% (over a baseline skill of 0.11). The average temporal skill over all forecast dates is displayed above each map.

As evidenced in Fig. 4, an important component of the overall accuracy of ABC is the reduction of the systematic bias introduced by dynamical model deficiencies. Figure 4 presents the spatial distribution of this bias by plotting the average difference between forecasts and observations over all forecast dates. The precipitation maps reveal a wet bias over the northern half of the U.S. for CFSv2 (average bias: 8.32 mm) and a dry bias over the south-east part of the U.S. for ECMWF (average bias: −8.12 mm). In this case, ABC eliminates the CFSv2 wet bias (average bias: −0.46 mm) and slightly alleviates the ECMWF dry bias (average bias: −6.24 mm). For temperature, we observe a cold bias over the eastern half of the U.S. for CFSv2 (average bias: −1.2 °C) and notice a mixed pattern of cold and warm biases over the western half of the U.S for ECMWF (average bias: −0.30 °C). In this case, although ABC does not eliminate these biases entirely, it reduces the magnitude of the cold eastern bias by bringing CFSv2 forecasts closer to observations (average bias: −0.18 °C) and reduces the mixed ECMWF bias (average bias: −0.04 °C).

We observe comparable bias reductions when ABC is applied to the SubX multimodel mean in Supplementary Fig. S11, improved dampening of bias relative to quantile mapping, LOESS, and neural network baselines in Supplementary Fig. S12, and improved dampening of bias relative to operationally debiased temperature and CFSv2 precipitation in Supplementary Fig. S13. Since each bias correction is based on historical data and weather is non-stationary, the remaining residual bias patterns may be indicative of recent regional shifts in average temperature or precipitation, e.g., decreased average precipitation in the Southeastern U.S. or increased average temperature on the country's coasts.

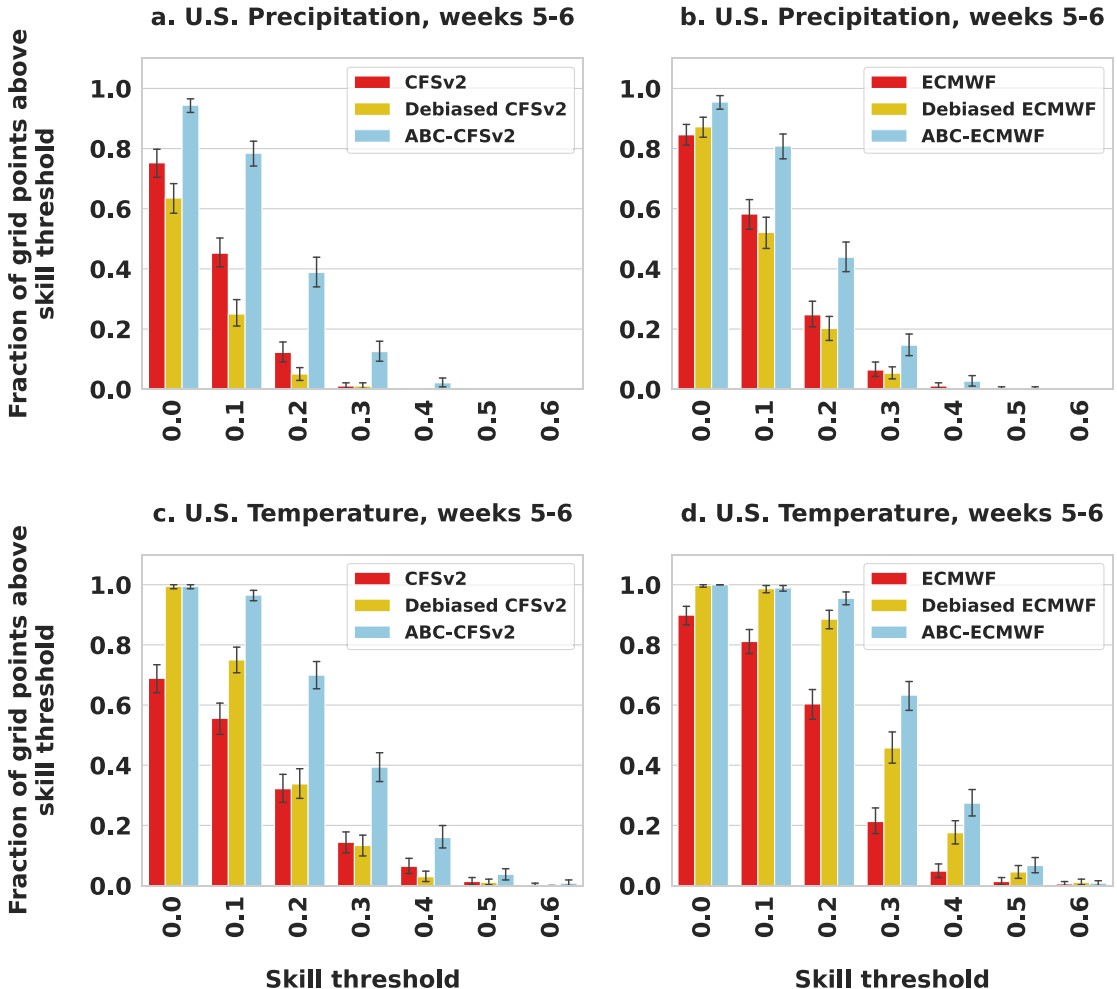

**Fig. 3 | Fraction of contiguous U.S. with 2018–2021 spatial skill above a given threshold.** For each forecasting task and dynamical model input (**a**–**d**), adaptive bias correction (ABC) consistently expands the geographic range of higher skill over raw and operationally debiased dynamical models. The error bars display 95% bootstrap confidence intervals.

## Identifying statistical forecasts of opportunity

The results presented so far evaluate overall model skill, averaged across all forecast dates. However, there is a growing appreciation that subseasonal forecasts can benefit from selective deployment during "windows of opportunity," periods defined by observable climate conditions in which specific forecasters are likely to have higher skill[5]. In this section, we propose a practical opportunistic ABC workflow that uses a candidate set of explanatory variables to identify windows in which ABC is especially likely to improve upon a baseline model. The same workflow can be used to explain the skill improvements achieved by ABC in terms of the explanatory meteorological variables.

The opportunistic ABC workflow is based on the equitable credit assignment principle of Shapley[33] and measures the impact of explanatory variables on individual forecasts using Cohort Shapley[31] and overall variable importance using Shapley effects[34] (see "Methods" section for more details). We use these Shapley measures to determine the contexts in which ABC offers improvements, in terms of climate variables with known relevance for subseasonal forecasting accuracy. As a running example, we use our workflow to explain the skill differences between ABC-ECMWF and debiased ECMWF when predicting precipitation in weeks 3–4. As our candidate explanatory variables, we use Northern Hemisphere geopotential heights (HGT) at 500 and 10 hPa, the phase of the MJO, Northern Hemisphere sea ice concentration, global sea surface temperatures, the multivariate El Niño–Southern Oscillation index (MEI.v2)[35], and the target month. All variables are lagged as described in "Methods" section to ensure that they are observable on the forecast issuance date.

We first use Shapley effects to determine the overall importance of each variable in explaining the precipitation skill improvements of ABC-ECMWF. As shown in Supplementary Fig. S14, the most important explanatory variables are the first two principal components (PCs) of 500 hPa geopotential height, the MJO phase, the second PC of 10 hPa geopotential height, and the first PC of sea ice concentration. These variables are consistent with the literature exploring the dominant contributions to subseasonal precipitation. The 500 hPa geopotential height plays a crucial role in conveying information about the thermal structure of the atmosphere and indicates synoptic circulation changes[36]. The MJO phase influences weather and climate phenomena within both the tropics and extratropics, resulting in a global influence of MJO in modulating temperature and precipitation[37]. The 10 hPa geopotential height is a known indicator of polar vortex variability leading to lagged impacts on sea level pressure, surface temperature, and precipitation[2]. Finally, sea ice concentration has a strong impact on surface turbulent heat fluxes and therefore near-surface temperatures[38].

We next use Cohort Shapley to identify the contexts in which each variable has the greatest impact on skill. For example, Fig. 5 summarizes the impact of the first 500 hPa geopotential heights PC

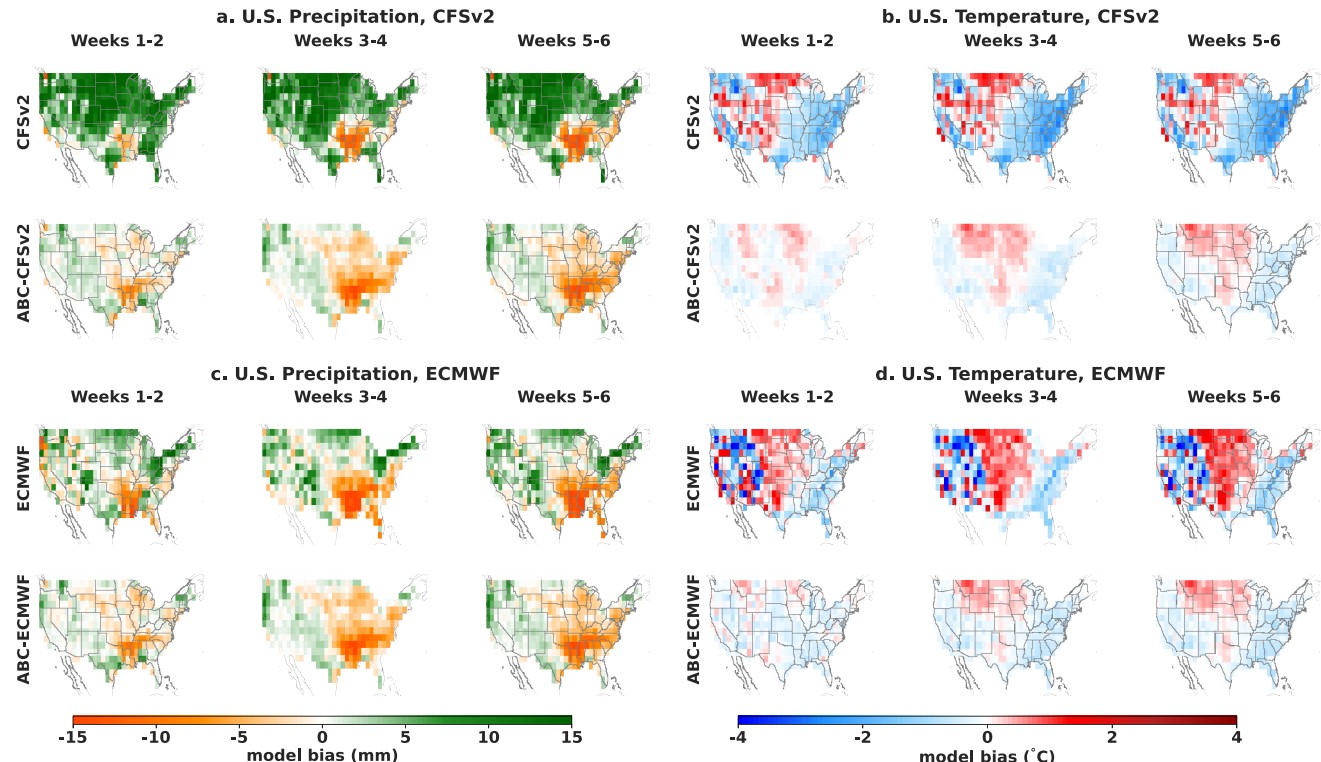

**Fig. 4 | Spatial distribution of model bias over the years 2018–2021.** Across the contiguous U.S., adaptive bias correction (ABC) reduces the systematic model bias of its dynamical model input for both precipitation (**a**, **c**) and temperature (**b**, **d**).

(hgt_500_pc1) on ABC-ECMWF skill improvement. This display divides our forecasts into 10 bins, determined by the deciles of hgt_500_pc1, and computes the probability of positive impact in each bin. We find that hgt_500_pc1 is most likely to have a positive impact on skill improvement in decile 1, which features a positive Arctic Oscillation (AO) pattern, and least likely in decile 9, which features AO in the opposite phase. The ABC-ECMWF forecast most impacted by hgt_500_pc1 in decile 1 is also preceded by a positive AO pattern and replaces the wet debiased ECMWF forecast with a more skillful dry pattern in the west. Similarly, Fig. 6 summarizes the impact of the MJO phase (mjo_phase) on ABC-ECMWF skill improvement. Importantly, while skill improvement is sometimes achieved with an especially skillful ABC forecast (as in Fig. 5), it can also be achieved by recovering from an especially poor baseline forecast. The latter is what we see at the bottom of Fig. 6, where the highest impact ABC forecast avoids the strongly negative skill of the baseline debiased ECMWF forecast.

Finally, we use the identified contexts to define windows of opportunity for operational deployment of ABC. Indeed, since all explanatory variables are observable on the forecast issuance date, one can selectively apply ABC when multiple variables are likely to have a positive impact on skill and otherwise issue a default, standard forecast (e.g., debiased ECMWF). We call this selective forecasting model opportunistic ABC. How many high-impact variables should we require when defining these windows of opportunity? We say a variable is "high-impact" if the positive impact probability for its decile or bin is within the confidence interval of the highest probability overall. Requiring a larger number of high-impact variables will tend to increase the skill gains of ABC but simultaneously reduce the number of dates on which ABC is deployed. Figure 7 illustrates this trade-off for ABC-ECMWF and shows that opportunistic ABC skill is maximized when two or more high-impact variables are required. With this choice, ABC is used for approximately 81% of forecasts and debiased ECMWF is used for the remainder. Figure 8 summarizes the complete opportunistic ABC workflow, from the identification of windows of

opportunity through the selective deployment of either ABC or a default baseline forecast for a given target date.

## Discussion

Dynamical models have shown increasing skill in accurately forecasting the weather[39], but they still contain systematic biases that compound on subseasonal timescales and suppress forecast skill[40–43]. ABC learns to correct these biases by adaptively integrating dynamical forecasts, historical observations, and recent weather trends. When applied to the leading subseasonal model from ECMWF, ABC improves forecast skill by 60–90% (over baseline skills of 0.18–0.25) for precipitation and 40–69% (over baseline skills of 0.11–0.15) for temperature. The same approach substantially reduces the forecasting errors of seven additional operational subseasonal forecasting models as well as their multimodel mean, with less skillful input models performing nearly as well as the ECMWF model after applying the ABC correction. This finding suggests that systematic errors in dynamical models are a primary contributor to observed skill differences and that ABC provides an effective mechanism for reducing these heterogeneous errors. Because ABC is also simple to implement and deploy in real-time operational settings, ABC represents a computationally inexpensive strategy for upgrading operational models, while conserving valuable human resources.

While the learned correction of systematic errors can play an important role in skill improvement, it is no substitute for scientific improvements in our understanding and representation of the processes underlying subseasonal predictability. As such, we view ABC as a complement for improved dynamical model development. Fortunately, ABC is designed to be adaptive to model changes. As operational models are upgraded, process models improve, and systematic biases evolve, our ABC training protocol is designed to ingest the upgraded model forecasts and hindcasts reflecting those changes.

To capitalize on higher-skill forecasts of opportunity, we have also introduced an opportunistic ABC workflow that explains the skill improvements of ABC in terms of a candidate set of environmental

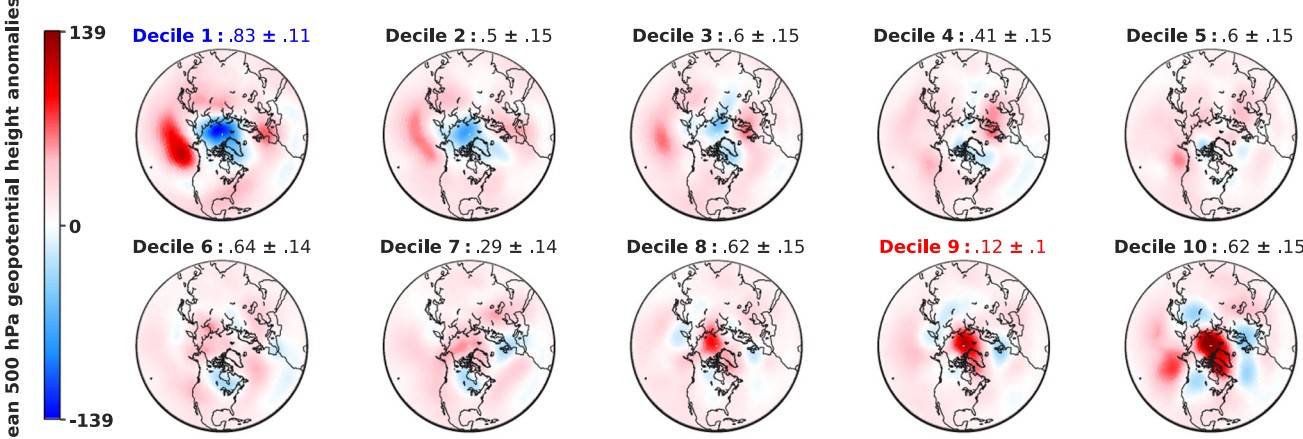

### a. Impact of hgt_500_pc1 on ABC-ECMWF skill for precipitation, weeks 3-4

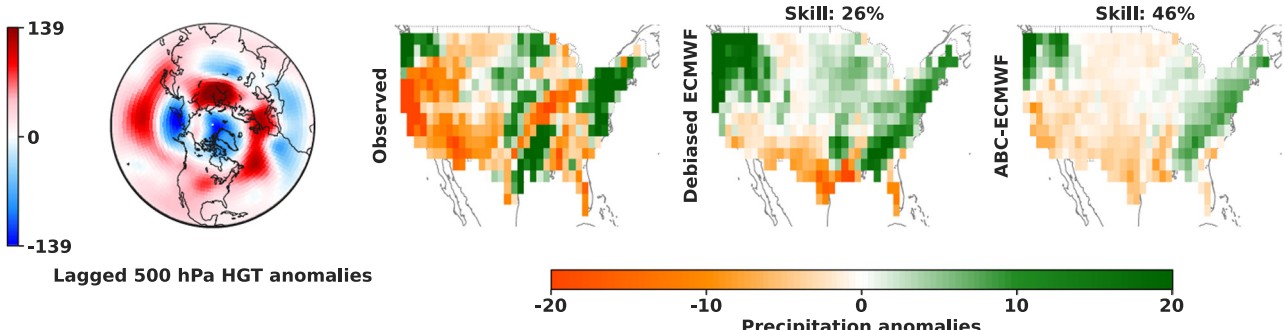

### b. Forecast with largest hgt_500_pc1 impact in decile 1: 2020-12-18

**Fig. 5 | Impact of the first 500 hPa geopotential heights principal component (hgt_500_pc1) on adaptive bias correction (ABC) skill improvement. a** To summarize the impact of hgt_500_pc1 on ABC-ECMWF skill improvement for precipitation weeks 3–4, we divide our forecasts into 10 bins, determined by the deciles of hgt_500_pc1, and display above each bin map the probability of positive impact in each bin along with a 95% bootstrap confidence interval. The highest probability of positive impact is shown in blue, and the lowest probability of

positive impact is shown in red. We find that hgt_500_pc1 is most likely to have a positive impact on skill improvement in decile 1 which features a positive Arctic Oscillation (AO) pattern, and least likely in decile 9, which features AO in the opposite phase. **b** The forecast most impacted by hgt_500_pc1 in decile 1 is also preceded by a positive AO pattern and replaces the wet debiased ECMWF forecast with a more skillful dry pattern in the west.

variables, identifies high-probability windows of opportunity based on those variables, and selectively deploys either ABC or a baseline forecast to maximize expected skill. The same workflow can be applied to explain the skill improvements of any forecasting model and, unlike other popular explanation tools[44,45], avoids expensive model retraining, requires no generation of additional forecasts beyond those routinely generated for operational or hindcast use, and allows for explanations in terms of variables that were not explicitly used in training the model.

Overall, we find that correcting dynamical forecasts using ABC yields an effective and scalable strategy to optimize the skill of the next generation of subseasonal forecasting models. We anticipate that our hybrid dynamical-learning framework will benefit both research and operations, and we release our open-source code to facilitate future adoption and development.

## Methods
### Dataset
All data used in this work was obtained from the SubseasonalClimateUSA dataset[46]. The spatial variables were interpolated onto a $1.5° × 1.5°$ latitude-longitude grid, and all daily observations (with two exceptions noted below) were aggregated into 2-week moving averages. As ground-truth measurements, we extracted daily gridded observations of average 2-meter

temperature in °C[47] and precipitation in mm[48–50]. For our explanatory variables, we obtained the daily PCs of 10 and 500 hPa stratospheric geopotential height[51] extracted from global 1948–2010 loadings, the daily PCs of sea surface temperature and sea ice concentration[52] using global 1981–2010 loadings, the daily MJO phase[53], and the bimonthly MEI.v2[35,54,55]. Precipitation was summed over two-week periods, and the MJO phase was not aggregated. Finally, we extracted twice-weekly ensemble mean forecasts of temperature and precipitation from the ECMWF S2S dynamical model[6] and ensemble mean forecasts of temperature and precipitation from seven models participating in the SubX project, including five coupled atmosphere-ocean-land dynamical models (NCEP-CFSv2, GMAO-GEOS, NRL-NESM, RSMAS-CCSM4, ESRL-FI) and two models with atmosphere and land components forced with prescribed sea surface temperatures (EMC-GEFS, ECCC-GEM)[11]. The SubX multimodel mean forecast was obtained by calculating, for each target date, the mean prediction over all available SubX models using the most recent forecast available from each model within a lookback window of size equal to 6 days. Each candidate SubX model is represented by its ensemble mean forecast. Two sets of candidate models are considered. When calculating SubX ensemble mean for weeks 1–2 and weeks 3–4, we consider NCEP-CFSv2, GMAO-GEOS, NRL-NESM, RSMAS-CCSM4, ESRL-FI, EMC-GEFS, and ECCC-GEM. When generating mean forecast for weeks 5–6, we

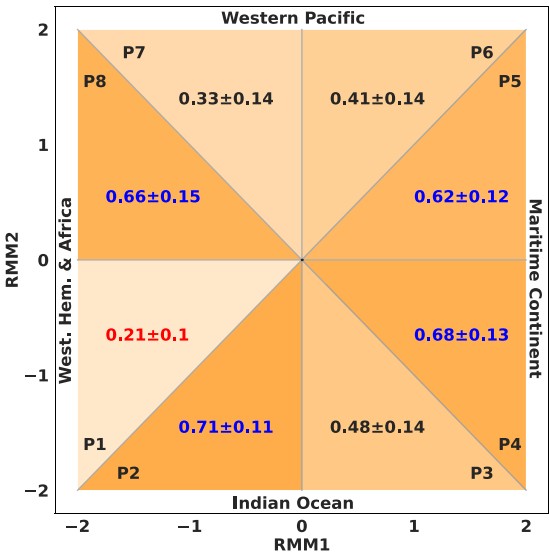

a. Impact of mjo_phase_lag17 on ABC-ECMWF skill for precipitation, weeks 3-4

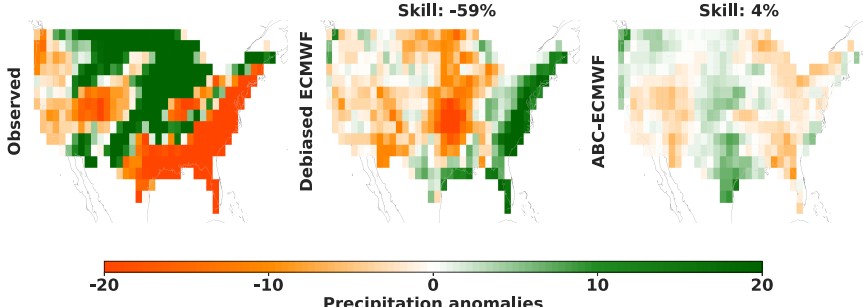

b. Forecast with largest mjo_phase_lag17 impact in phases 2, 4, 5, 8: 2019-09-24

**Fig. 6 | Impact of the Madden–Julian Oscillation phase (**mjo_phase**) on adaptive bias correction (ABC) skill improvement. a** To summarize the impact of mjo_phase on ABC-ECMWF skill improvement for precipitation weeks 3–4, we compute the probability of positive impact and an associated 95% bootstrap confidence interval in each lagged MJO phase bin and adopt the methodology of ref. [53] to create an MJO phase space diagram. The highest probabilities of positive impact (those falling within the confidence interval of the highest probability overall) are shown in blue and the lowest probability of positive impact is shown in red. We find that positive impact on skill improvement is most common in phases 2, 4, 5, and 8 and least common in phase 1. **b** The forecast most impacted by mjo_phase in phases 2, 4, 5, and 8 avoids the strongly negative skill of the debiased ECMWF baseline.

consider NCEP-CFSv2, GMAO-GEOS, NRL-NESM, and RSMAS-CCSM4 only, as the remaining models do not produce forecast data for weeks 5–6.

### Forecasting tasks and skill assessment

We consider two prediction targets: average temperature (°C) and accumulated precipitation (mm) over a 2-week period. These variables are forecasted at two time horizons: 15–28 days ahead (weeks 3–4) and 29–42 days ahead (weeks 5–6). We forecast each variable at $G = 376$ grid points on a 1.5° × 1.5° grid across the contiguous U.S., bounded by latitudes 25N–50N and longitudes 125W–67W. To provide the most realistic assessment of forecasting skill[56], all predictions in this study are formed in a real forecast manner that mimics operational use. In particular, to produce a forecast for a given target date, all learning-based models are trained and tuned only on data observable on the corresponding forecast issuance date.

For evaluation, we adopt the exact protocol of the recent Sub-seasonal Climate Forecast Rodeo competition, run by the U.S. Bureau of Reclamation in partnership with the National Oceanic and Atmospheric Administration, U.S. Geological Survey, U.S. Army Corps of Engineers, and California Department of Water Resources[57]. In particular, for a 2-week period starting on date $t$, let $\mathbf{y}_t \in \mathbb{R}^G$ denote the vector of ground-truth measurements $y_{t,g}$ for each grid point $g$ and

$\hat{\mathbf{y}}_t \in \mathbb{R}^G$ denote a corresponding vector of forecasts. In addition, define climatology $\mathbf{c}_t$ as the average ground-truth values for a given month and day over the years 1981–2010. We evaluate each forecast using uncentered anomaly correlation skill[57,58],

$$\text{skill}(\hat{\mathbf{y}}_t, \mathbf{y}_t) = \frac{\langle \hat{\mathbf{y}}_t - \mathbf{c}_t, \mathbf{y}_t - \mathbf{c}_t \rangle}{\| \hat{\mathbf{y}}_t - \mathbf{c}_t \|_2 \cdot \| \mathbf{y}_t - \mathbf{c}_t \|_2} \in [-1,1], \tag{1}$$

with a larger value indicating higher quality. For a collection of target dates, we report average skill using progressive validation[59] to mimic operational use.

### Operational ECMWF, CFSv2, and SubX debiasing

We bias correct a uniformly weighted ensemble of the ECMWF control forecast and its 50 ensemble forecasts following the ECMWF operational protocol[60]: for each target forecast date and grid point, we bias correct the 51-member ensemble forecast by subtracting the equal-weighted 11-member ECMWF ensemble reforecast averaged over all dates from the last 20 years within ±6 days from the target month-day combination and then adding the average ground-truth measurement over the same dates.

Following ref. [57], we bias correct a uniformly weighted 32-member CFSv2 ensemble forecast, formed from four model

## a. Number of high-impact variables

| # High-impact variables | % Forecasts using ABC | High-impact skill (%) | |
|---|---|---|---|
| | | ABC | Debiased |
| 0 or more | 100.00 | 20.94 | 15.28 |
| 1 or more | 95.93 | 20.99 | 14.84 |
| 2 or more | 80.62 | 22.29 | 13.12 |
| 3 or more | 58.61 | 23.56 | 12.00 |
| 4 or more | 31.82 | 24.72 | 8.18 |
| 5 or more | 14.59 | 26.51 | 8.35 |
| 6 or more | 6.46 | 29.72 | 10.55 |
| 7 or more | 2.15 | 45.00 | 17.53 |

## b. Opportunistic ABC skill

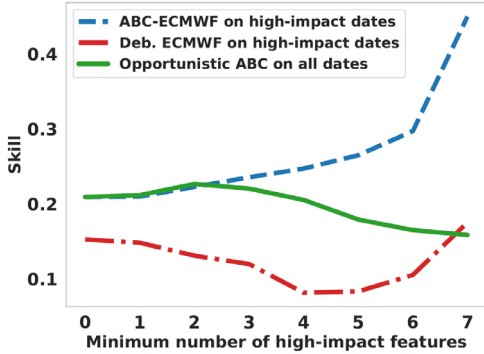

**Fig. 7 | Defining windows of opportunity for opportunistic adaptive bias correction (ABC) forecasting.** Here we focus on forecasting precipitation in weeks 3–4. **a** When more explanatory variables fall into high-impact deciles or bins (e.g., the blue bins of Figs. 5 and 6), the mean skill of ABC-ECMWF improves, but the percentage of forecasts using ABC declines. **b** The overall skill of opportunistic ABC is maximized when ABC-ECMWF is deployed for target dates with two or more high-impact variables and standard debiased ECMWF is deployed otherwise.

initializations averaged over the eight most recent 6-hourly issuances, in the following way: for each target forecast date and grid point, we bias correct the 32-member ensemble forecast by subtracting the equal-weighted 8-member CFSv2 ensemble hindcast averaged over all dates from 1999 to 2010 inclusive matching the target day and month and then adding the average ground-truth measurement over the same dates. We bias correct the SubX multimodel mean forecast in an identical manner, using the SubX multimodel mean reforecasts from 1999 to 2010 inclusive.

### Adaptive bias correction

ABC is a uniformly weighted ensemble of three machine learning models, Dynamical++, Climatology++, and Persistence++, detailed below. A schematic of ABC model input and output data can be found in Supplementary Fig. S15, and supplementary algorithm details can be found in Supplementary Methods.

Dynamical++ (Algorithm S1) is a three-step approach to dynamical model correction: (i) adaptively select a window of observations around the target day of year and a range of issuance dates and lead times for ensembling based on recent historical performance, (ii) form an ensemble mean forecast by averaging over the selected range of issuance dates and lead times, and (iii) bias correct the ensemble forecast for each site by adding the mean value of the target variable and subtracting the mean forecast over the selected window of observations. Unlike standard debiasing strategies, which employ static ensembling and bias correction, Dynamical++ adapts to heterogeneity in forecasting error by learning to vary the amount of ensembling and the size of the observation window over time.

For a given target date $t^\star$ and lead time $l^\star$, the Dynamical++ training set $\mathcal{T}$ is restricted to data fully observable one day prior to the issuance date, that is, to dates $t \leq t^\star - l^\star - L - 1$ where $L = 14$ represents the forecast period length. For each target date, Dynamical++ is run with the hyperparameter configuration that achieved the smallest mean progressive geographic root mean squared error (RMSE) over the preceding 3 years. Here, progressive indicates that each candidate model forecast is generated using all training data observable prior to the associated forecast issuance date. Every configuration with span $s \in \{0, 14, 28, 35\}$ (the span is the number of days included on each side of the target day of year), number of averaged issuance dates $d^\star \in \{1, 7, 14, 28, 42\}$, and leads $\mathcal{L} = \{29\}$ for the weeks 5–6 lead time and $\mathcal{L} \in \{\{15\}, [15,22], [0,29], \{29\}\}$ the weeks 3–4 lead time was considered.

Inspired by climatology, Climatology++ (Algorithm S2) makes no use of the dynamical forecast and rather outputs the historical geographic median (if the user-supplied loss function is RMSE) or mean (if

loss = MSE) of its target variables over all days in a window around the target day of year. Unlike a static climatology, Climatology++ adapts to target variable heterogeneity by learning to vary the size of the observation window and the number of training years over time.

For a given target date $t^\star$ and lead time $l^\star$, the Climatology++ training set $\mathcal{T}$ is restricted to data fully observable one day prior to the issuance date, that is, to dates $t \leq t^\star - l^\star - L - 1$ where $L = 14$ represents the forecast period length. For each target date, Climatology++ is run with the hyperparameter configuration that achieved the smallest mean progressive geographic RMSE over the preceding 3 years. All spans $s \in \{0, 1, 7, 10\}$ were considered. All precipitation configurations used the geographic MSE loss and all available training years. All temperature configurations used the geographic RMSE loss and either all available training years or $Y = 29$. For shorter than subseasonal lead times (e.g., weeks 1–2), Climatology++ is excluded from the ABC forecast and only Dynamical++ and Persistence++ are averaged.

Persistence++ (Algorithm S3) accounts for recent weather trends by fitting an ordinary least-squares regression per grid point to optimally combine lagged temperature or precipitation measurements, climatology, and a dynamical ensemble forecast. For a given target date $t^\star$ and lead time $l^\star$, the Persistence++ training set $\mathcal{T}$ is restricted to data fully observable one day prior to the issuance date, that is, to dates $t \leq t^\star - l^\star - L - 1$ where $L = 14$ represents the forecast period length. In Algorithm S3, the set $\mathcal{L}$ represents the full set of subseasonal lead times available in the dataset, i.e., $\mathcal{L} = [0, 29]$.

### Debiasing baselines

NN-A[22] learns a non-linear mapping between daily corrected CFSv2 precipitation and temperature and observed precipitation and temperature for the contiguous U.S. In particular, the model's inputs (predictors) consist of CFSv2 bias-corrected ensemble mean for total precipitation and temperature anomalies as well as the observed climatologies for precipitation and temperature. The model's target variables (predictands) are observed temperature anomalies and total precipitation. Both daily target variables are converted to 2-weekly mean and 2-weekly total and are predicted simultaneously for the entire forecast domain. NN-A is a neural network with a single hidden layer consisting of $K = 200$ hidden neurons. This architecture enables the model to account for both non-linear relationships among input and target variables as well as their spatial dependency and the co-variability that characterize these variables. For a given lead time, the NN-A model was trained on all available data from January 2000 to December 2017 (inclusive) save for those dates that were unobservable on the issuance date associated with a January 1, 2018, target date. Each

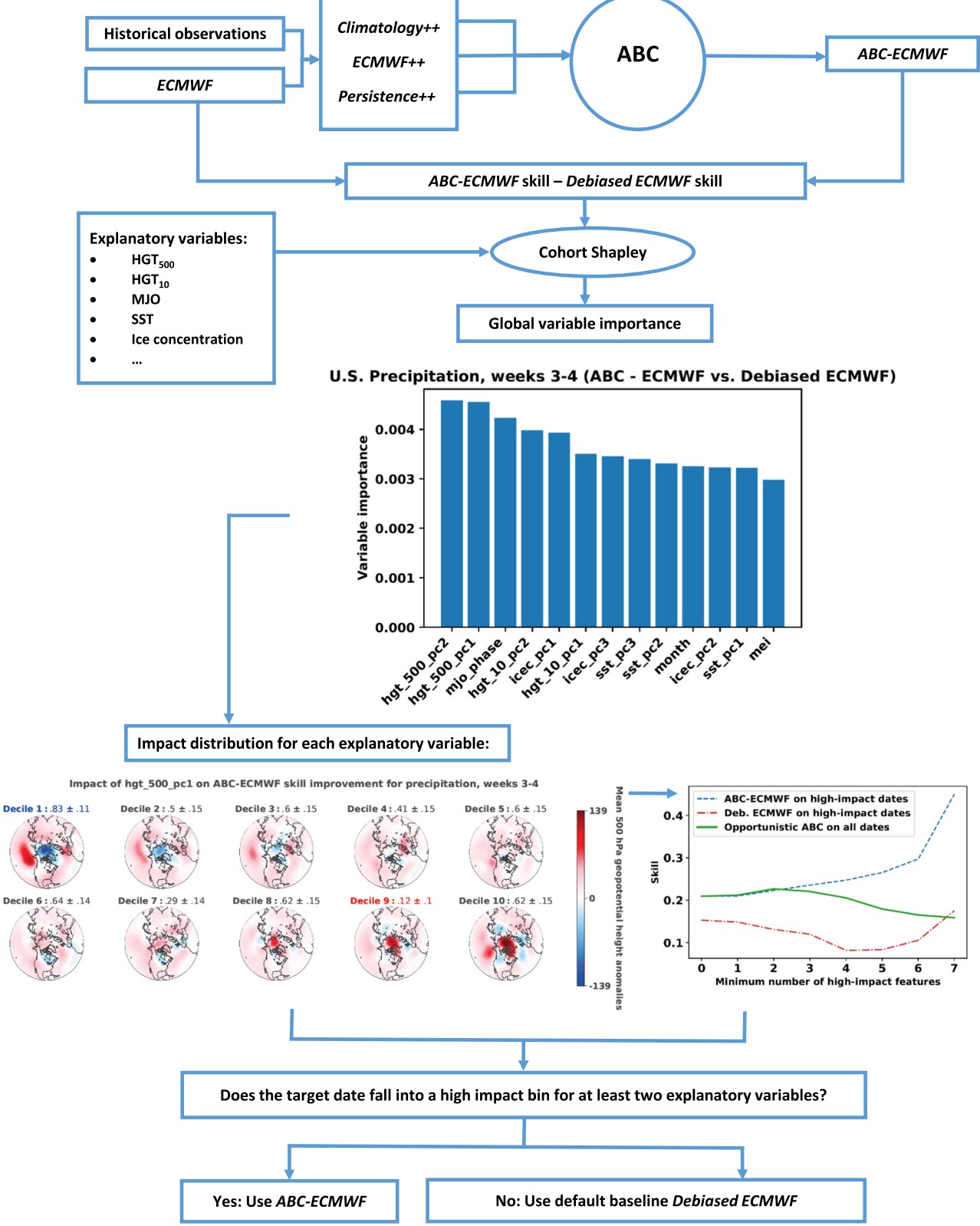

**Fig. 8 | Schematic of the opportunistic adaptive bias correction (ABC) workflow.** Opportunistic ABC uses historical ABC and baseline forecasts and a candidate set of explanatory variables to identify windows of opportunity for selective deployment of ABC in an operational setting.

NN-A model was trained using the Adam algorithm[61] for 10,001 epochs without dropout as in ref. 22 and used relu activations and the default batch size (32) and learning rate (0.001) from Tensorflow[62].

LOESS debiasing[30] adds a correction to a dynamical model forecast using LOESS. Using all dates prior to 2018 with available ground-truth measurements and (re)forecast data, the measurements for each month-day combination (save February 29) are averaged, resulting in a sequence of 365 values. The same is done for the forecasts. A local linear regression is run on each of these sequences using a fraction of 0.1 of the points to fit each value. The end result is two smoothed

sequences of 365 values, one with the measurement data and the other one with forecast data. The entrywise difference (in the case of temperature) or ratio (in the case of precipitation) between these sequences is used as a correction to be added (in the case of temperature) or multiplied (in the case of precipitation) to the forecasts made in 2018 and beyond, based on the target forecast day and month. Note that the locality of the smoothed corrections, which only use consecutive days in the calendar year and do not wrap around from December to January, ensures that every forecast is made using only training data observable on the forecast issuance date.

Quantile mapping[26] corrects a base dynamical forecast by aligning the quantiles of forecast and measurement data. For our training set, we use all dates prior to 2018 with available ground-truth measurements and (re)forecast data. For a given grid point, target date, and dynamical model forecast, we first identify the quantile rank of the forecasted value amongst all training set forecasts issued for the same month-day combination. If the quantile rank exceeds 90%, we replace its value with 90%; if the quantile rank falls below 10%, we replace its value with 10%. We then add to the forecast the corresponding quantile of the training set measurements for the target month and day and subtract the corresponding quantile of the training set forecasts for the target month and day. In the case of precipitation, if the resulting value is negative, we set the forecast to zero.

### Cohort Shapley and Shapley effects

Cohort Shapley and Shapley effects use Shapley values to quantify the impact of variables on outcomes. Shapley values are based on work in game theory[33] exploring how to assign appropriate rewards to individuals who contribute to an outcome as part of a team. When applied to explanatory variables, Shapley values can be thought of as roughly analogous to the coefficients in a linear regression. Importantly, unlike linear regression coefficients, Shapley values are applicable in settings where the interaction among variables is highly non-linear. The procedure for computing Shapley values involves testing how much a change to one explanatory variable influences a target outcome. These tests are carried out by measuring how the target outcome varies when a given explanatory variable changes in the context of subsets of other explanatory variables.

Shapley effects[34] are a specific instantiation of the general Shapley value principle, designed for measuring variable importance. For a given outcome variable to be explained (for example, the skill difference between ABC-ECMWF and operationally debiased ECMWF measured on each forecast date) and a collection of candidate explanatory variables (for example, relevant meteorological variables observed at the time of each forecast's issuance), the Shapley effects are overall measures of variable importance that quantify how much of the outcome variable's variance is explained by each candidate explanatory variable. Cohort Shapley values[31] provide a more granular application of Shapley values by quantifying the impact of each explanatory variable on the measured outcome of each individual forecast.

### Opportunistic ABC workflow

Here we detail the steps of the opportunistic ABC workflow illustrated in Fig. 8 using ECMWF as an example of dynamical input. The same workflow applies to any other dynamical input.

1. Identify a set of V candidate explanatory variables. Here we use the temporal variables enumerated in ref. 15 (Fig. 2) augmented with the first two PCs of 500 hPa geopotential heights and the target month. To ensure that the workflow can be deployed operationally, we use lagged observations with lags chosen so that each variable is observable on the forecast issuance date. MEI.v2 is lagged by 45 days when forecasting weeks 3–4 and by 59 days for weeks 5–6. The other variables are lagged by 30 days for weeks 3–4 and by 44 days for weeks 5–6.

2. Compute the temporal skill difference between ABC-ECMWF and debiased ECMWF for each target date in the evaluation period.

3. For each continuous explanatory variable (e.g., hgt_500_pc2), divide the evaluation period forecasts into 10 bins, determined by the deciles of the explanatory variable. For each categorical variable (e.g., mjo_phase), divide the forecasts into bins determined by the categories (e.g., MJO phases).

4. Use the `cohortshapley` Python package to compute overall variable importance (measured by Shapley effects) and forecast-specific variable impact values explaining the skill differences.

5. Within each variable bin, compute the fraction of forecasts with positive Cohort Shapley impact values. Report that fraction as an estimate of the probability of positive variable impact, and compute a 95% bootstrap confidence interval. Flag all bin probabilities within the confidence interval of the highest probability bin as high impact; similarly, flag all bin probabilities within the confidence interval of the lowest probability bin as low impact. The remaining bins—those that fall outside of both confidence intervals—have an intermediate impact and are not flagged as either low or high impact. Visualize and interpret the highest and lowest impact bins.

6. Identify the forecast most impacted by the explanatory variable in the high-impact bins. Visualize the ABC-ECMWF and debiased ECMWF forecasts and the associated explanatory variable for that target date.

7. For each $k \in \{0, ..., V\}$, compute opportunistic ABC skill when $k$ or more explanatory variables fall into high-impact bins. Let $k^\star$ represent the integer at which opportunistic ABC skill is maximized.

8. At each future forecast issuance date, deploy ABC-ECMWF if $k^\star$ or more explanatory variables fall into high-impact bins and deploy debiased ECMWF otherwise.

## Data availability

The SubseasonalClimateUSA dataset used in this study has been deposited on Microsoft Azure and is available for download via the `subseasonal_data` Python package: https://github.com/microsoft/subseasonal_data. This work is based on S2S data. S2S is a joint initiative of the World Weather Research Programme (WWRP) and the World Climate Research Programme (WCRP). The original S2S database is hosted at ECMWF as an extension of the TIGGE database. We acknowledge the agencies that support the SubX system, and we thank the climate modeling groups (Environment Canada, NASA, NOAA/NCEP, NRL and University of Miami) for producing and making available their model output. NOAA/MAPP, ONR, NASA, NOAA/NWS jointly provided coordinating support and led the development of the SubX system. Source Data are provided with this paper.

## Code availability

Python 3 code replicating all experiments and analyses in this work is available at https://github.com/microsoft/subseasonal_toolkit.

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

## Acknowledgements

We thank Jessica Hwang for her suggestion to explore Cohort Shapley. This work was supported by Microsoft AI for Earth (S.M. and G.F.); the Climate Change AI Innovation Grants program (S.M., P.O., G.F., J.C., E.F., and L.M.), hosted by Climate Change AI with the support of the Quadrature Climate Foundation, Schmidt Futures, and the Canada Hub of Future Earth; FAPERJ (Fundação Carlos Chagas Filho de Amparo à Pesquisa do Estado do Rio de Janeiro) grant SEI-260003/001545/2022 (P.O.); NOAA grant OAR-WPO-2021-2006592 (G.F., J.C., and L.M.); and the National Science Foundation grant PLR-1901352 (J.C.).

## Author contributions

S.M., P.O., G.F., J.C., E.F., and L.M. designed the research, discussed the results, and contributed to the writing of the manuscript. S.M., P.O., G.F., M.O., and L.M. performed model runs. S.M., P.O., and L.M. analyzed model outputs and generated figures.

## Competing interests

The authors declare no competing interests.
