## [Peer Review File · Nature Communications]

Adaptive Bias Correction for Improved Subseasonal ForecastingREVIEWER COMMENTS

Reviewer #1 (Remarks to the Author):

Adaptive Bias Correction for Improved Subseasonal Forecasting

Soukayna M. et al.

Adaptive Bias Correction (or ABC) is proposed as a hybrid dynamical-statistical method that can significantly improve the skill of operational subseasonal forecasts of temperature and precipitation. ABC is an ensemble of three machine learning models trained on past observational data, forecasts data and outputs corrections that are location/time/model specific. A related practical framework based on Cohort Shapley is proposed to explain the gains and identify higher-skill windows of opportunity. All this is proposed for operational use as an enhancement to dynamical models. All the code is made publicly available. Results are provided for the 2018-2021 period, independent of the data used for training and corrections, and applied to forecasts of temperature and precipitation for weeks 1-2, 3-4 and 5-6.

Although the results are technical and of direct applicability to the forecasting community, the problem is broadly important and such operational skill gains are well sought after for practical purposes.

Overall the methodology is well-explained and convincing as a post-processing approach that can provide incremental yet useful increases to the skill by addressing some model biases. The explanatory framework, attributing the skill gains to a particular variable or phenomena, and also to help identify windows of opportunity, is interesting.

My main criticism is that the paper should more centrally acknowledge that model biases are one factor limiting the forecast skill but not the only one. In fact, even when ABC (or debiasing) is applied to fix some of these issues, the skill is still quite modest. This points to the need to further understand processes underpinning predictability e.g., the role of initial conditions and process representation, and how to evolve prediction system design to take full advantage of processes that underpin predictability. Post-processing methodologies cannot fix some of these fundamental issues. For this reason, the paper needs to provide a more balanced view regarding the value of post-processing to remedy forecast limitations and also discuss predictability science for more significant breakthroughs. Speaking to the subseasonal problem as a "predictability desert" as in some of the older references that were provided, is a bit outdated, as we now understand much more about predictability sources.

A recommendation would be to add the results for ABC applied to the SubX mean and not just to each SubX model. We know that model mean typically has better performance than that of any individual model. Does ABC improve upon the SubX model mean?

There are a number of specific issues and suggestions, as listed below.

Specific Comments

- For clarity, replace "these two extremes" with "beyond 10 days and up to a season".
- Predictability desert? Actually, we now know that several processes are at play at sub-seasonal skill. We still need to work on how to optimally represent these processes in our models, and also figure out whether there are other unknown processes at play. The paper should spend a bit more time discussing predictability sources in addition to talking about model biases.
- "This scale is incompatible.." It is not just a question of model resolution but also process understanding and representation. Resolution is important but previous work has shown that it is not going to fix all issues.
- "These biases..and translate into rapidly decreasing skill for precipitation forecasts". There is also a question of chaotic dynamics and predictability. It is not just model biases. While model biases certainly influence skill, removing biases does not automatically translate into perfect skill scores - results in this paper show just that.
- Figure 1. Can one see the results for the mean of the SubX models in this and other spatial

pattern figures?

-The paper discusses skill gains in terms of %. This is not the best way to represent increases given that the original skill is in many cases close to zero.

-The statement that skill improvement is due to bias decrease should point to Figure 3 otherwise it is unsubstantiated.

-Figure 2. There seem to be some features common across models. Any discussion here?

-Figure 3. Similarly, there are bias patterns that are not eliminated. Any discussion of what this means?

-The paper did not offer results stratified by season. I assume this is because of the limited time record but there was no discussion. We know that not all seasons are alike predictability-wise so it would be an interesting follow-up.

-The paper could do with a careful re-read. For instance all references to the Supplementary material say "??". It is surprising that authors submitted a paper to a Nature journal without double checking all cross references.

-References seem to be a bit old. Other relevant work on the topic should be acknowledged, e.g. <https://www.nature.com/articles/s41467-021-23406-3>

Reviewer #3 (Remarks to the Author):

Review of manuscript NCOMMS-22-38161 entitled “Adaptive Bias Correction for Improved Subseasonal Forecasting” by Soukayna Mouatadid, Paulo Orenstein, Genevieve Flaspohler, Judah Cohen, Miruna Oprescu, Ernest Fraenkel and Lester Mackey

OVERALL RECOMMENDATION

Major revision

SUMMARY

This manuscript proposes an advanced bias correction methodology for dynamical subseasonal forecasts of temperature and precipitation over CONUS, called Adaptive Bias Correction (ABC). In this study, ABC combines three equally-weighted machine learning algorithms. The authors’ results tend to show that ABC enhances the skill of the ECMWF S2S and SubX forecasts at lead times weeks 3-4 and 5-6, compared to raw and baseline debiased forecasts. Moreover, the authors propose a methodology to analyze which explanatory variables play an important role in the improvement, as well as a workflow to determine in real-time whether a forecaster should use the baseline or ABC debiasing procedure.

MAJOR COMMENTS

This study features two important topics for research on subseasonal forecasting: statistical post-processing using machine learning and real-time identification of windows of opportunity. The various results suggest the manuscript can be a significant contribution to the field. However, I think it cannot be published before some major improvements in terms of clarity, especially in the description of the methodology. My overall feeling is that of a “black box”: the authors provide interesting results, but it is hard to grasp the path that leads to them.

The main issues towards reproducibility that I see are the following:

1) The description of the three ABC machine learning models is really unclear, in spite of Supplementary information #2. I think the overall philosophy of each model should be summarized with a sentence, and that some figure showing the data which is fed to each model would really help. It is particularly important to make clear how these three models differ from a standard bias correction.

2) The Shapley framework is quite new in atmospheric sciences, so I think an additional description of the main principles is necessary. The authors should also clearly connect those general principles to their specific case (e.g what are the predictors, the predicted

values, the black box model). For instance, I still struggle to understand where the probabilities for each decile come from in Figure 4.

3) “All variables are lagged appropriately to ensure that they are observable on the forecast issuance date”: could you specify what is the lag between the variable and the initialization date?

MINOR COMMENTS

1) Figure 1: I assume that some models disappear between the above panels (weeks 3-4) and the below panels (weeks 5-6) because they do not forecast up to that lead time. I think this should appear in the caption.

2) Figure 3: What about the bias patterns for baseline debiased forecasts, as in Figure 2? I assume they are non-zero since debiasing is performed with reforecasts. Then it would be interesting to see whether the ABC method is really better at dampening the bias patterns than standard debiasing.

3) Figure 6, caption and the associated paragraph in the text: The authors specify that high-impact variables are variables falling into “the blue bins of Figure 4 and 5”. However, they should also add the proper definition, i.e. determined from the bootstrap procedure as described in the workflow below.

4) Figure 7: I guess that the difference “ABC-ECMWF skill – Debiased ECMWF skill” is a spatial average over all grid points, but it could be specified somewhere for the sake of clarity.

5) Could the authors briefly state why they choose to assess skill with uncentered anomaly correlation instead of standard anomaly correlation which is far more common? I guess the aim is to have a fair comparison between the three different kinds of forecasts (raw, standard debiasing and ABC) but this could be explicitly specified.

6) Being color-blind, I would recommend to avoid some color palettes such as the one the authors use in all maps referring to precipitation (Figure 3 left, Figure 4 bottom, Figure 5 bottom). I am personally not able to make a distinction between the grid points corresponding to the opposite sides of the palette. I suggest to use either a red/blue palette (the same as for temperature and Z500) or another combination with strong contrast (in my experience vivid orange and dark green works well).

7) Opportunistic ABC workflow, step #5:

“Flag all bin probabilities **within** the confidence interval of the highest prob-

ability bin as high impact and all bin probabilities **within** the confidence interval of the lowest probability bin as low impact.”

This sentence is nonsensical and confusing as “within” is repeated twice. One of them should be replaced by an expression such as “out of the confidence interval”. If I have understood well, it should be the first one.

8) I am not familiar with the Nature Communications layout so please ignore this comment if irrelevant, but I find it strange that scientific comments appear in the figures captions (e.g Figure 1 “ABC provides a pronounced improvement in skill for each forecasting task”).

9) There are lots of minor issues in the LaTeX compilation of the PDF files, e.g references as ?? and bad display of bibliographical references. Please also check that your references are complete (e.g missing URL for [6], missing journal for [51]).

Reviewer #3 (Remarks to the Author):

The manuscript proposes a bias correction technique for forecasts at the s2s time scale. The adaptive bias correction technique presented is interesting and novel.

The main concerns I have are related to the scoring metrics used and the lack of bias correction baselines presented in this manuscript. The paper presents a new bias correction method for leading NWP models without offering a sufficient review of current bias correction literature and how they perform compared to the new proposed technique. It is difficult to assess the value of the proposed new technique without any baselines. There is a fairly large body of literature on machine learning based bias correction methods at this point so one would expect to have a brief review and comparisons to prior work when reading about a new ML bias correction method.

My second issue is about the scoring metrics used. At S2S timescales, I would expect to have some probabilistic skill scores reported along with the ACC of weekly average. I would expect some attempt at reporting a probabilistic skill score like CRPS or brier scores at some thresholds for precipitation or at the very least some discussion regarding probabilistic skill.

Finally, there are a lot of broken references in the methods section of the paper. This is not a major issue but it should have been fixed before submission.

In summary, I believe this paper, while interesting and technically correct, does not rise to the threshold of novelty and completeness for publication in Nature communications.

Response to Reviewers

We are encouraged by the supportive comments regarding our manuscript, and we thank the reviewers for their detailed and constructive feedback. In our revision, we have attempted to address all suggestions and concerns, and we provide point-by-point responses below.

Reviewer 1

Reviewer Point P 1.1 — My main criticism is that the paper should more centrally acknowledge that model biases are one factor limiting the forecast skill but not the only one. In fact, even when ABC (or debiasing) is applied to fix some of these issues, the skill is still quite modest. This points to the need to further understand processes underpinning predictability e.g., the role of initial conditions and process representation, and how to evolve prediction system design to take full advantage of processes that underpin predictability. Post-processing methodologies cannot fix some of these fundamental issues. For this reason, the paper needs to provide a more balanced view regarding the value of post-processing to remedy forecast limitations and also discuss predictability science for more significant breakthroughs. Speaking to the subseasonal problem as a "predictability desert" as in some of the older references that were provided, is a bit outdated, as we now understand much more about predictability sources.

Reply: Thank you for this important note on the messaging of the paper, both with regards to how predictability on subseasonal timescales is discussed and on the importance of improving process modeling versus post-processing correction. We have adjusted this messaging in several places in the paper:

1. The second paragraph of the introduction now includes a discussion on the sources of predictability for subseasonal timescales.
2. The third paragraph of the introduction now highlights process understanding, chaotic dynamics, and imperfectly represented sources of predictability as sources of error.
3. The sixth paragraphs of the introduction now centrally acknowledges that correction alone is no substitute for improved understanding and representation of predictability sources and that we view ABC as a complement for improved dynamical modeling.
4. Similarly, the second paragraph of the discussion now centrally acknowledges that learned correction is no substitute for scientific improvements in our understanding and representation of the processes underlying subseasonal predictability and that ABC should be viewed as a complement for improved dynamical model development.

Reviewer Point P 1.2 — A recommendation would be to add the results for ABC applied to the SubX mean and not just to each SubX model. We know that model mean typically has better performance than that of any individual model. Does ABC improve upon the SubX model mean?

Reply: Thank you for this recommendation. We now include the SubX multimodel mean in Figure 1, display SubX multimodel mean spatial skill distributions in Supplementary Figure A3, and present

the spatial distribution of SubX multimodel mean bias in Supplementary Figure A9. In each case, we observe improvements from ABC comparable to those previously established for the individual dynamical models.

Reviewer Point P 1.3 — For clarity, replace “these two extremes” with “beyond 10 days and up to a season”.

Reply: Thank you; we have implemented this suggestion in the introduction.

Reviewer Point P 1.4 — Predictability desert? Actually, we now know that several processes are at play at sub-seasonal skill. We still need to work on how to optimally represent these processes in our models, and also figure out whether there are other unknown processes at play. The paper should spend a bit more time discussing predictability sources in addition to talking about model biases.

Reply: We have added information about the sources of predictability to the introductory text and modified the language about the “predictability desert”; please see the response to P 1.1 as well.

Reviewer Point P 1.5 — “This scale is incompatible..” It is not just a question of model resolution but also process understanding and representation. Resolution is important but previous work has shown that it is not going to fix all issues.

Reply: Thank you for raising this issue which we have addressed in concert with P 1.1 in the Introduction.

Reviewer Point P 1.6 — “These biases..and translate into rapidly decreasing skill for precipitation forecasts”. There is also a question of chaotic dynamics and predictability. It is not just model biases. While model biases certainly influence skill, removing biases does not automatically translate into perfect skill scores - results in this paper show just that.

Reply: We have changed this language about precipitation forecasting to acknowledge the role of chaos and predictability and to emphasize that the goal is not perfect prediction but instead getting closer to estimated predictability limits. We have also acknowledged more explicitly that bias correction should be pursued in concert with process and model improvement; please see the response to P 1.1 as well.

Reviewer Point P 1.7 — Figure 1. Can one see the results for the mean of the SubX models in this and other spatial pattern figures?

Reply: Please see our reply to P 1.2.

Reviewer Point P 1.8 — The paper discusses skill gains in terms of %. This is not the best way to represent increases given that the original skill is in many cases close to zero.

Reply: Thank you for this feedback. We now report baseline skills alongside any percentage skill gains so that the reader has complete information.

Reviewer Point P 1.9 — -The statement that skill improvement is due to bias decrease should point to Figure 3 otherwise it is unsubstantiated.

Reply: Thank you; we have implemented this suggestion in Section 2.

Reviewer Point P 1.10 — -Figure 2. There seem to be some features common across models. Any discussion here?

Reply: Thank you for this suggestion; we have added a discussion of these common features across models to the Results section (see Section 2).

Reviewer Point P 1.11 — -Figure 3. Similarly, there are bias patterns that are not eliminated. Any discussion of what this means?

Reply: Thank you for this suggestion; we have added a discussion of what these residual bias patterns might indicate to the Results section (see Section 2).

Reviewer Point P 1.12 — -The paper did not offer results stratified by season. I assume this is because of the limited time record but there was no discussion. We know that not all seasons are alike predictability-wise so it would be an interesting follow-up.

Reply: Following this recommendation, we have now added forecast skill results stratified by season to Supplementary Appendix A1. In each season, we observe ABC skill improvements analogous to the overall improvements displayed in Figure 1.

Reviewer Point P 1.13 — -The paper could do with a careful re-read. For instance all references to the Supplementary material say “?”. It is surprising that authors submitted a paper to a Nature journal without double checking all cross references.

Reply: We apologize for the unresolved supplement references which arose when we separated the supplement from the main paper at the time of submission. We have implemented a workaround that restores these references and have conducted a careful reread of all submitted materials.

Reviewer Point P 1.14 — -References seem to be a bit old. Other relevant work on the topic should be acknowledged, e.g. <https://www.nature.com/articles/s41467-021-23406-3>

Reply: Thank you for this recommendation. We now cite the following additional references from the recent subseasonal forecasting literature: [Monhart et al., 2018, Baker et al., 2019, Li et al., 2019, Fan et al., 2021, Kim et al., 2021].

Reviewer 2

Reviewer Point P 2.1 — The description of the three ABC machine learning models is really unclear, in spite of Supplementary information #2. I think the overall philosophy of each model should be summarized with a sentence, and that some figure showing the data which is fed to each model would really help. It is particularly important to make clear how these three models differ from a standard bias correction.

Reply: Thank you for these recommendations. To improve clarity, we have updated the initial descriptions of the three ABC models in Section 4 as follows and added an schematic of ABC model input and output data as Supplementary Figure A13.

- **Dynamical++** is a three-step approach to dynamical model correction: (i) adaptively select a window of observations around the target day of year and a range of issuance dates and lead times for ensembling based on recent historical performance, (ii) form an ensemble mean forecast by averaging over the selected range of issuance dates and lead times, and (iii) bias-correct the ensemble forecast for each site by adding the mean value of the target variable and subtracting the mean forecast over the selected window of observations. Unlike standard debiasing strategies, which employ static ensembling and bias correction, Dynamical++ adapts to heterogeneity in forecasting error by learning to vary the amount of ensembling and the size of the observation window over time.
- Inspired by climatology, **Climatology++** makes no use of the dynamical forecast and rather outputs the historical geographic median (if the user-supplied loss function is RMSE) or mean (if loss = MSE) of its target variables over all days in a window around the target day of year. Unlike a static climatology, Climatology++ adapts to target variable heterogeneity by learning to vary the size of the observation window and the number of training years over time.
- **Persistence++** accounts for recent weather trends by fitting an ordinary least-squares regression per grid point to optimally combine lagged temperature or precipitation measurements, climatology, and a dynamical ensemble forecast.

Reviewer Point P 2.2 — The Shapley framework is quite new in atmospheric sciences, so I think an additional description of the main principles is necessary. The authors should also clearly connect those general principles to their specific case (e.g what are the predictors, the predicted values, the black box model). For instance, I still struggle to understand where the probabilities for each decile come from in Figure 4.

Reply: Thank you for this recommendation. We have now added two paragraphs to Section 4 describing the principles underlying Shapley’s fair allocation framework, Shapley effects, and Cohort Shapley and how those principles translate concretely into our setting. In addition, step 5 in the “Opportunistic ABC workflow” subsection of Methods (see Section 4) now clarifies that the positive variable impact probability for each decile is estimated as the fraction forecasts belonging to that decile that have positive impact values assigned by Cohort Shapley.

Reviewer Point P 2.3 — “All variables are lagged appropriately to ensure that they are observable on the forecast issuance date”: could you specify what is the lag between the variable and the initialization date?

Reply: Yes, we have added these details to Section 4.

Reviewer Point P 2.4 — Figure 1: I assume that some models disappear between the above panels (weeks 3-4) and the below panels (weeks 5-6) because they do not forecast up to that lead time. I think this should appear in the caption.

Reply: Your understanding is correct, and we have incorporated this suggestion into Figure 1.

Reviewer Point P 2.5 — Figure 3: What about the bias patterns for baseline debiased forecasts, as in Figure 2? I assume they are non-zero since debiasing is performed with reforecasts. Then

it would be interesting to see whether the ABC method is really better at dampening the bias patterns than standard debiasing.

Reply: Following this suggestion, we have added two new figures comparing the bias of ABC to that of operational debiasing and several additional debiasing baselines based on quantile mapping, LOESS, and neural networks, respectively. We observe improved dampening of bias relative to quantile mapping, LOESS, and neural network baselines in Supplementary Figure A10 and improved dampening of bias relative to operationally debiased temperature and CFSv2 precipitation in Supplementary Figure A11.

Reviewer Point P 2.6 — Figure 6, caption and the associated paragraph in the text: The authors specify that high-impact variables are variables falling into “the blue bins of Figure 4 and 5”. However, they should also add the proper definition, i.e determined from the bootstrap procedure as described in the workflow below.

Reply: Thank you for this suggestion. We now clarify the definition of “high-impact” in the paragraph referencing Figure 6.

Reviewer Point P 2.7 — Figure 7: I guess that the difference “ABC-ECMWF skill – Debiased ECMWF skill” is a spatial average over all grid points, but it could be specified somewhere for the sake of clarity.

Reply: We now clarify, in step 2 of the “Opportunistic ABC workflow” segment of the Methods section (see Section 4), that this difference refers to temporal skill rather than spatial skill.

Reviewer Point P 2.8 — Could the authors briefly state why they choose to assess skill with uncentered anomaly correlation instead of standard anomaly correlation which is far more common? I guess the aim is to have a fair comparison between the three different kinds of forecasts (raw, standard debiasing and ABC) but this could be explicitly specified.

Reply: In Section 4, we now clarify that we adopted the exact evaluation protocol of the Subseasonal Climate Forecast Rodeo competition, run by the U.S. Bureau of Reclamation in partnership with the National Oceanic and Atmospheric Administration, U.S. Geological Survey, U.S. Army Corps of Engineers, and California Department of Water Resources [Nowak et al., 2017]. In particular, to provide a standardized metric for benchmarking subseasonal forecasting methods, these agencies adopted uncentered anomaly correlation as their measurement of skill, and we use the same skill measure for our evaluation.

Reviewer Point P 2.9 — Being color-blind, I would recommend to avoid some color palettes such as the one the authors use in all maps referring to precipitation (Figure 3 left, Figure 4 bottom, Figure 5 bottom). I am personally not able to make a distinction between the grid points corresponding to the opposite sides of the palette. I suggest to use either a red/blue palette (the same as for temperature and Z500) or another combination with strong contrast (in my experience vivid orange and dark green works well).

Reply: We apologize and have now changed the precipitation maps to use the recommended vivid orange vs. dark green palette.

Reviewer Point P 2.10 — Opportunistic ABC workflow, step #5: “Flag all bin probabilities within the confidence interval of the highest probability bin as high impact and all bin probabilities within the confidence interval of the lowest probability bin as low impact.” This sentence is nonsensical and confusing as “within” is repeated twice. One of them should be replaced by an expression such as “out of the confidence interval”. If I have understood well, it should be the first one.

Reply: The use of “within” twice in this sentence is intentional, as the high impact probabilities lie in an interval around the **highest** probability bin, and the low impact probabilities lie in an interval around the **lowest** probability bin. That is, there are two separate confidence intervals being discussed, one for flagging especially high impact bins and one for flagging especially low impact bins. To drive home this point, we have added the following sentence to the description in Section 4: “The remaining bins – those that fall outside of both confidence intervals – have intermediate impact and are not flagged as either low or high impact.”

Reviewer Point P 2.11 — I am not familiar with the Nature Communications layout so please ignore this comment if irrelevant, but I find it strange that scientific comments appear in the figures captions (e.g Figure 1 “ABC provides a pronounced improvement in skill for each forecasting task”).

Reply: We have reviewed the journal policies and believe our captions are in compliance, but we would be happy to edit the captions if we are mistaken.

Reviewer Point P 2.12 — There are lots of minor issues in the LaTeX compilation of the PDF files, e.g references as ?? and bad display of bibliographical references. Please also check that your references are complete (e.g missing URL for [6], missing journal for [51]).

Reply: We apologize for the unresolved supplement references which arose when we separated the supplement from the main paper at the time of submission. We have implemented a workaround that restores these references. In addition, we have repaired the two mentioned references and reviewed the remaining references for completeness.

Reviewer 3

Reviewer Point P 3.1 — The main concerns I have are related to the scoring metrics used and the lack of bias correction baselines presented in this manuscript. The paper presets a new bias correction method for leading NWP models without offering a sufficient review of current bias correction literature and how they perform compared to the new proposed technique. It is difficult to assess the value of the proposed new technique without any baselines. There is a fairly large body of literature on machine learning based bias correction methods at this point so one would expect to have a brief review and comparisons to prior work when reading about a new ML bias correction method.

Reply: Thank you for this feedback and recommendation. We now highlight several learned bias corrections adopted in the recent subseasonal forecasting literature in the introduction (see Section 1), describe each of these baselines in the Methods section (see Section 4), and compare ABC to each

baseline in terms of average forecast skill (Supplementary Figure A2), the spatial distribution of model skill (Supplementary Figure A4), and the spatial distribution of model bias (Supplementary Figure A10). We find that, given the same dynamical model inputs, ABC provides improved skill and dampened bias relative to each baseline for each forecasting task.

Reviewer Point P 3.2 — My second issue is about the scoring metrics used. At S2S timescales, I would expect to have some probabilistic skill scores reported along with the ACC of weekly average. I would expect some attempt at reporting a probabilistic skill score like CRPS or brier scores at some thresholds for precipitation or at the very least some discussion regarding probabilistic skill.

Reply: Thank you for this recommendation. We now clarify in Section 2 that, like each of the aforementioned subseasonal baselines [Monhart et al., 2018, Baker et al., 2019, Li et al., 2019, Fan et al., 2021] and recent subseasonal forecasting work published in Nature Communications [e.g., Kim et al., 2021], our results focus on improved deterministic forecasting. We then discuss the complementary paradigm of probabilistic forecasting and highlight that we would normally advocate for a distinct and more tailored approach to probabilistic debiasing that directly optimizes a probabilistic skill measure to output a corrected distribution. Nevertheless, we describe a simple, inexpensive way to convert the output of ABC into a probabilistic forecast and show in Supplementary Figures A5 and A6 that, for each target variable, forecast horizon, and season, this strategy leads to improved continuous ranked probability scores (CRPS) and Brier skill scores (BSS) for the leading subseasonal forecasting model from ECMWF. In Supplementary Figures A7 and A8 we further observe that ABC-corrected ECMWF improves upon the CRPS and BSS of baseline LOESS and quantile mapping corrections.

Reviewer Point P 3.3 — Finally, there are a lot of broken references in the methods section of the paper. This is not a major issue but it should have been fixed before submission.

Reply: We apologize for the unresolved supplement references which arose when we separated the supplement from the main paper at the time of submission. We have implemented a workaround that restores these references.

References

- S. A. Baker, A. W. Wood, and B. Rajagopalan. Developing subseasonal to seasonal climate forecast products for hydrology and water management. *JAWRA Journal of the American Water Resources Association*, 55(4):1024–1037, 2019.
- Y. Fan, V. Krasnopolsky, H. van den Dool, C.-Y. Wu, and J. Gottschalck. Using artificial neural networks to improve CFS week 3-4 precipitation and 2-meter air temperature forecasts. *Weather and Forecasting*, 2021. doi: 10.1175/WAF-D-20-0014.1. URL <https://journals.ametsoc.org/view/journals/wefo/aop/WAF-D-20-0014.1/WAF-D-20-0014.1.xml>.
- H. Kim, Y. Ham, Y. Joo, and S. Son. Deep learning for bias correction of mjo prediction. *Nature communications*, 12(1):1–7, 2021.
- W. Li, J. Chen, L. Li, H. Chen, B. Liu, C.-Y. Xu, and X. Li. Evaluation and bias correction of S2S precipitation for hydrological extremes. *Journal of Hydrometeorology*, 20(9):1887–1906, 2019.

- S. Monhart, C. Spirig, J. Bhend, K. Bogner, C. Schär, and M. A. Liniger. Skill of subseasonal forecasts in Europe: Effect of bias correction and downscaling using surface observations. *Journal of Geophysical Research: Atmospheres*, 123(15):7999–8016, 2018.
- K. Nowak, R. Webb, R. Cifelli, and L. Brekke. Sub-seasonal climate forecast rodeo. In *2017 AGU Fall Meeting*, pages 11–15, New Orleans, LA, 2017.

REVIEWERS' COMMENTS

Reviewer #1 (Remarks to the Author):

The paper is much improved in view of the authors' revisions. Many of my previous points have been addressed.

A few additional points below.

1. It's good to see absolute baseline skill along side % values. However, this highlights that the skill scores are still very low, even after correction. The authors should add a discussion of the practical value of these corrected skill score i.e. whether there is any.
2. "In addition, precipitation is oftentimes a very local phenomenon, working over a much finer scale than the resolution employed by subseasonal dynamical models and therefore not fully resolved in global climate models." The discussion here is about sub-seasonal to seasonal dynamical models and forecasts, not about climate models. Hence, remove "and therefore not fully resolved in global climate models" which is out of scope.
3. LOESS is introduced and only much later in the paper spelled out. Please double check that the acronyms for the various methods are spelled out.

Reviewer #2 (Remarks to the Author):

The authors have satisfactorily addressed my remarks. I think this article is fit for publication.

Reviewer #3 (Remarks to the Author):

I appreciate the authors' responses and thank them for their revisions. The revisions are reasonable and address most of my concerns. I lean towards a recommendation to accept the manuscript for publication with moderate confidence.

Response to Reviewers

We once again thank the reviewers for their positive and constructive feedback. In our revision, we have attempted to address all remaining comments, and we provide point-by-point responses below.

Reviewer 1

Reviewer Point P 1.1 — The paper is much improved in view of the authors' revisions. Many of my previous points have been addressed.

Reply: Thank you!

Reviewer Point P 1.2 — It's good to see absolute baseline skill along side % values. However, this highlights that the skill scores are still very low, even after correction. The authors should add a discussion of the practical value of these corrected skill score i.e. whether there is any.

Reply: Thank you for this suggestion. We have now added the following discussion of the practical value of the ABC corrected skills to the main manuscript (along with the referenced Figure 3 and supplementary Figures S5 and S6 summarizing the expanded ranges of actionable skill):

“A practical implication of these improvements for downstream decision-makers is an expanded geographic range for actionable skill, defined here as spatial skill above a given sufficiency threshold. For example, in Figure 3, we vary the weeks 5-6 sufficiency threshold from 0 to 0.6 and find that ABC consistently boosts the number of locales with actionable skill over both raw and operationally-debiased CFSv2 and ECMWF. We observe similar gains for weeks 3-4 in Figure S5 and for ABC correction of the SubX multimodel mean in Figure S6.”

Reviewer Point P 1.3 — “In addition, precipitation is oftentimes a very local phenomenon, working over a much finer scale than the resolution employed by subseasonal dynamical models and therefore not fully resolved in global climate models.” The discussion here is about sub-seasonal to seasonal dynamical models and forecasts, not about climate models. Hence, remove “and therefore not fully resolved in global climate models” which is out of scope.

Reply: We have now removed “and therefore not fully resolved in global climate models.”

Reviewer Point P 1.4 — LOESS is introduced and only much later in the paper spelled out. Please double check that the acronyms for the various methods are spelled out.

Reply: We now spell out LOESS when it is first used and have checked that other acronyms are also spelled out.

Reviewer 2

Reviewer Point P 2.1 — The authors have satisfactorily addressed my remarks. I think this article is fit for publication.

Reply: Thank you!

Reviewer 3

Reviewer Point P 3.1 — I appreciate the authors' responses and thank them for their revisions. The revisions are reasonable and address most of my concerns. I lean towards a recommendation to accept the manuscript for publication with moderate confidence.

Reply: Thank you!

References